# Impact of Pathway Shape and Length on the Validity of the 6-Minute Walking Test: A Systematic Review and Meta-Analysis

**DOI:** 10.3390/s25010017

**Published:** 2024-12-24

**Authors:** Armelle-Myriane Ngueleu, Solène Barrette, Coralie Buteau, Chloé Robichaud, Michelle Nguyen, Gauthier Everard, Charles Sebiyo Batcho

**Affiliations:** 1Centre for Interdisciplinary Research in Rehabilitation and Social Integration, Quebec City, QC G1M 2S8, Canada; armelle-myriane.ngueleu.1@ulaval.ca (A.-M.N.); michelle.nguyen.3@ulaval.ca (M.N.); gauthier-jean-marc-a.everard.1@ulaval.ca (G.E.); 2School of Rehabilitation Sciences, Faculty of Medicine, Université Laval, Quebec City, QC G1V 0A6, Canada; solene.barrette.1@ulaval.ca (S.B.); coralie.buteau.1@ulaval.ca (C.B.); chloe.robichaud.1@ulaval.ca (C.R.); 3Faculty of Medicine, Université Laval, Quebec City, QC G1V 0A6, Canada

**Keywords:** functional assessment, walking test, walked distance, psychometric, validity

## Abstract

Although guidelines are established for performing the six-minute walking test (6MWT), it is not always possible to implement this test in any setting, due to physical and space limitations. Yet, variations in the conditions of the test could be responsible for heterogeneous outcomes. However, the impact of the condition of the 6MWT is not clearly established in literature. The objective is to determine the influence of different implementation conditions on the validity of the 6MWT. Seventeen articles were retained after a literature review, including 597 participants. Seven articles mention that performing the test on a predetermined short back-and-forth pathway led to lower performance than when the test was performed on pathways of greater distances. The walking distance covered on a rectangular path or on a 10-m eight-form path is greater than with the back and forth on a five- to ten-meter path. Seven articles suggest that the performance achieved on a treadmill is generally lower than that obtained while walking on the ground. Evidence shows that the conditions while performing the 6MWT significantly influence the score, hence the validity of the results. The use of a ground pathway, comprising the longest linear distance possible, seems critical to ensure good validity.

## 1. Introduction

Physiotherapists must constantly assess and reassess different parameters involving their patients in their practice. A common functional assessment of patients is important and even recommended to define goals and treatment strategy and to determine recovery. For this purpose, there are many measurement tools and functional tests, such as the six-minute walking test (6MWT). The 6MWT is a functional test that is frequently used in rehabilitation as a performance-based measure of functional exercise capacity. Developed in 1963 to assess the functional capacity in elderly people, this test is now used with a wide range of populations as an assessment and monitoring tool [1]. The test consists of walking for six minutes on a predefined pathway. The patient is instructed to walk at the maximal safe speed. Distance traveled is measured and compared with previous outcome or the guidelines established by previous studies and corresponding to the population to which the patient belongs [2]. Expected distance on the 6MWT depends on the population as well as on an individual’s age, weight, sex, and height. For example, a healthy man aged 75 years with 85 Kg of weight and 175 cm of height should travel 616.81 m of distance on the 6MWT [3]. In the normative studies, the 6MWT is generally performed following the guidelines of the American Thoracic Society (ATS), which recommend performing the test on a straight 30 m pathway where the patient walks back and forth for six minutes [2]. However, it is sometimes difficult to respect these recommendations in a clinical setting, due to physical and space limitations, and few studies report on the impact of performing the 6MWT in other conditions. Thus, it is not clear how different implementation conditions influence the validity of the 6MWT results.

The first hypothesis is that the shorter the straight-line distance traveled back and forth by the participants is, the smaller the total walked distance will be. Indeed, performing the 6MWT on a shorter pathway requires more frequent 180-degree turns. The second hypothesis is that performing the 6MWT on a treadmill should impact balance, as walking on a treadmill requires more dynamic motor control [4]. The third hypothesis is that subjects who perform the test on a pathway with different shapes and walking back and forth (ex. path in a circle, oval, or rectangle) would obtain a more representative result of their functional capacity compared to a 6MWT performed with traditional back and forth since they would not have to reduce their speed as much to do a 180-degree turn.

In this review, it is considered that the condition in which the total distance is the longest is the most valid since the subjects can travel a distance that best represents their maximum functional capacity. Therefore, according to the hypotheses above, the validity of the results obtained with the 6MWT would be better when performing the 6MWT on shapes such as rectangles, ovals, and circles than on a course with many back and forth motions or on a treadmill.

This systematic review identifies and summarizes literature on the impact of the shape, length, and surface pathway on the validity of the 6MWT scores within different populations in order to highlight evidence that could guide clinical practice and then provide appropriate interpretations of functional exercise capacity.

## 2. Materials and Methods

This systematic review was prospectively registered with the International Prospective Register of Systematic Reviews (PROSPERO) in November 2023 (CRD42023468405).

### 2.1. Scientific Research and Article Identification

Publications were searched for in three databases, Medline, CINAHL, and Embase, updated on 3 January 2024, to identify the studies that assessed the impact of the test’s implementation conditions (shape, ground or treadmill, and length of pathway) on 6MWT score. Keywords were selected based on three different concepts: (1) the 6MWT, (2) implementation condition, and (3) psychometric properties of the test. Finally, these three concepts were merged and separated by “AND” to obtain the final search equation. For each database, the keywords were adapted to respect the truncations and the controlled terms (MeSH, describer CINAHL, and Emtree). The search terms are presented in Appendix A.

### 2.2. Inclusion and Exclusion Criteria for Article Selection

To be included, articles had (1) to report comparative data on the 6MWT performed with at least two different conditions related to either the shape or length of the pathway or the use of a treadmill, (2) to assess the validity of the 6MWT according to the different implementation conditions (shape, length, and surface pathways), and (3) to be published in French or English. Conference abstracts, systematic reviews, and meta-analyses were excluded.

### 2.3. Articles Selection

Articles were selected using the Covidence platform following the “Preferred Reporting Items for Systematic Reviews and Meta-Analyses” (PRISMA) guidelines [5]. First, two reviewers initially screened the articles by reading only the title and the abstract to determine whether they were potentially eligible. In the absence of consensus, a third reviewer was called upon to participate. Afterwards, the full texts of articles deemed to be eligible were independently read by two of the three reviewers who participated in the initial screening for a decision regarding inclusion or exclusion based on predefined criteria. If one of them was excluded, the reviewer was required to record the associated exclusion criteria. In the case of a conflict, the third reviewer was involved in the final decision of inclusion or exclusion. In addition, references of selected articles were checked for additional articles to be included.

### 2.4. Data Extraction

Two reviewers participated in the data extraction for each included article. The first reviewer collected data, and the second reviewer verified the extracted data and completed the data where needed. The items extracted for this systematic review were the article title, year of publication, type of study, specific characteristics of studied population (number of participants, age, sex, presence or not of an associated disability), the 6MWT conditions (shape, length, and surface pathway), the compliance with the ATS recommendations related to the encouragements given to the subjects, the distance traveled at the 6MWT, the assessment context (clinical or research), and the study’s conclusion.

### 2.5. Assessment of the Methodological Quality

The methodological quality of the articles included in this review was assessed with the Strengthening the Reporting of Observational Studies in Epidemiology (STROBE) tool. This is a checklist of 22 items, including sub-items, and divided according to the different sections of an article (title, abstract, introduction, methodologies, results, discussion, and other information (funding)) [6,7]. An independent reviewer scored each sub-item according to its presence or absence in the article with “yes” (Y), “no” (N), or “non applicable” (NA). Given the lack of standardized scores available for STROBE, the number of “yes” votes was counted for each sub-item and reported on the total of applicable items and sub-items, for a maximum of 34 points. The results were then presented in percentages to compare the quality of the different articles. An article that yields a score of ≥80.0% is considered to have excellent quality. A score between 70.0 and 79.9% is considered good quality. A score between 50.0 and 69.9% is considered moderate quality, and <50.0% is considered poor quality [8,9].

### 2.6. Statistical Analyses

Forest plots were generated using RevMan 5.3 software. Number of participants, mean distance (in meters) covered and associated standard deviation were extracted for each condition: traditional 6MWT, treadmill, home-based, 20 m pathway, 15 m pathway, 10 m pathway, 5 m pathway, figure eight pathway, rectangle pathway, and oval pathway versions. All information was entered into the software, and a mean difference (MD) with a 95% confidence interval was computed to compare the distance covered during the traditional 6MWT to that of every other condition. According to the Cochrane Handbook, “the MD is a standard statistic that measures the absolute difference between the mean value in two groups or conditions in a trial. It estimates the amount by which the experimental intervention/condition changes the outcome on average compared with the control/other condition. It can be used as a summary statistic in meta-analysis when outcome measurements in all studies are made on the same scale.” Between-studies heterogeneity was assessed using the I² test. According to the Cochrane Handbook, heterogeneity was rated as non-significant when I^2^ < 30%, moderate for an I^2^ between 30% and 50%, substantial for an I^2^ between 50% and 75%, and considerable when I^2^ > 75%.

Statistical significance was defined as a *p*-value < 0.05. The *p*-value, ranging from 0 to 1, is the probability of observing a result as extreme as, or more extreme than, the observed one, assuming the null hypothesis is true. It is directly related to the risk of making a type I error, which occurs when the null hypothesis is incorrectly rejected.

## 3. Results

Following the literature search in databases, 5503 articles were identified after 1160 duplicates were removed. The titles and abstracts screening led to the exclusion of 4317 records, and 26 articles were retained for full text assessment, where 11 other articles were excluded. Indeed, one article was written in Spanish [10], two were not available in full text [11,12], one did not report relevant results to the purpose of this systematic review [13], and seven articles did not compare the shape or the length of the 6MWT pathway [11,14,15,16,17,18,19]. Finally, 15 articles were selected, and two other articles that were identified from reference checking and determined as relevant were manually added to this review, for a total of 17 included articles (see Figure 1).

### 3.1. Characteristics of Included Studies

The influence of the length of the pathway with back-and-forth walking was assessed in 47.05% (8/17) of the articles [15,20,21,22,23,24,25,26]. Then, the performance of the 6MWT on a treadmill was compared to performance while walking in a corridor (with turn) in 7 [8,27,28,29,30,31,32] out of the 17 articles included. Finally, the influence of the shape of the pathway with no back-and-forth walking (oval, circle, rectangle, figure eight, or eight form) was assessed in 17.64% (3/17) of the articles [21,33,34]. Of the 17 articles, two [23,31] did not use the standardized directives by the ATS. Eleven studies [21,22,24,25,26,27,28,29,31,32,33] were conducted in a research context, while the six others [8,15,20,23,30,34] were conducted in a clinical context (see Table 1).

### 3.2. Participant’s Description

Within the 17 articles, a total of 597 participants were assessed, with sample sizes ranging from 10 to 109 participants [26,31]. Participants were 21 to 87 years old, refs. [20,33] and included 338 men and 259 women. In five articles, the population consisted solely of healthy participants [22,25,28,29,33] (see Table 2).

In total, 11 articles (64.70%) described how they calculated the sample size or the rationale behind their sampling method [8,15,21,22,23,24,25,28,29,31,34]. Among them, Cheng and et al.’s [15] article was the only one that did not recruit enough participants according to their calculations; nevertheless, this method did not affect their STROBE score. Also, six articles (35.29%) did not report sample size calculation [20,26,27,30,32,33], with three having a small number of participants (<25 participants) [32,33]. Finally, more than half of the included studies (58.82%) used appropriate sample sizes in their study [8,20,21,22,23,24,25,27,28,29,30,31,34].

### 3.3. Articles’ Quality and Methodology

The overall methodological quality scores ranged from 34.5% to 100% [15,33]. In fact, Cheng and et al.’s article [15] mentions that the study reported was written according to the STROBE guidelines, which explains its 100% score. Overall, the included studies presented a moderate methodological quality of around 57%. However, six articles that were not cross-sectional studies and where study design was not mentioned obtained a low methodological quality score of less than or equal to 50% [8,26,27,29,32,33]. It is important to note that no articles were excluded from the systematic review because of their STROBE scores (see Table 3).

### 3.4. Impact of Condition

As shown in Table 4, in eight articles, the distance walked during the 6MWT was shorter when the test was performed with back and forth on short distances (515.2 m) than long distances (544.0 ± 72.1 m) [15,20,22,23,24,25,26]. Among those studies, four reported a significant difference [15,20,21,23,25], and one reported a significant difference only for the subjects who scored more than 430 m [24]. One article reported that performing the 6MWT on a circular pathway produced variable results depending on the population [34]. In addition, another article mentioned that performing the 6MWT in a rectangular or an eight-form pathway resulted in a greater walking distance compared to the back-and-forth pathway of less than 15 m. One article reported that performing the 6MWT on a 10 m eight-form pathway yielded a shorter walking distance than the standard 30 m back-and-forth pathway [21]. However, when compared to the back-and-forth pathway of more than 15 m, the walking distance obtained was less [33]. Finally, seven articles [8,27,28,29,30,31,32] mentioned that the scores of the 6MWT performed on a treadmill were lower than those obtained by walking in a hallway; four of them strongly supported this conclusion [27,28,31,32]. According to Liu and et al.’s study [31], the results were significantly lower on a treadmill compared to walking on the ground for older participants with chronic obstructive pulmonary disease (COPD). However, the results were significantly higher on the treadmill compared with walking on the ground for healthy elderly people. Second, three articles [8,29,30] also determined that performing the test on a treadmill yielded lower results than walking on the floor, but it was not significant. They determined that both conditions were not exchangeable, but each were individually valid.

According to our meta-analysis and as presented in Figure 2, the distance covered during the traditional 6MWT was not significantly lower when using a treadmill (mean difference—MD = 43.16 [0.85 to 85.48]; *p* = 0.05). However, this analysis suffered from considerable heterogeneity (I² = 87%). A neutral difference was also observed for the home-based version of the test (MD = 30 [−38.57 to 98.57]; *p* = 0.39). Contrastingly, all 20 m (MD = 19.17 [0.99 to 37.35]; *p* = 0.04), 15 m (MD = 38.37 [16.16 to 61.57]; *p* < 0.001), 10 m (MD = 69.42 [53.37 to 85.47]; *p* < 0.001), and 5 m (MD = 155.84 [133.38 to 178.31]; *p* < 0.001) versions of the 6MWT offered a significantly lower distance when compared the traditional test (*p* < 0.05). As underlined by Figure 2, the smaller the pathway, the greater the gap with the distance covered during the traditional 6MWT. Regarding the shape of the pathway, the mean distances covered during both figure eight (MD = 54.26 [34.18 to 74.34]; *p* < 0.001) and rectangle pathways (MD = 92.60 [67.21 to 117.99]; *p* < 0.001) were significantly lower than those of the traditional 6MWT. The oval version observed no significant difference with the traditional test (MD = 26.90 [−14.26 to 68.06]; *p* = 0.20).

## 4. Discussion

The goal of this systematic review was to compare the influence of the implementation condition on the validity of the 6MWT score. Included studies had analyzed three factors, namely the length of the hallway in which participants walked back and forth (from 5 to 30 m), the type of surface (ground or treadmill), and the shape of the pathway (rectangle, circle, or eight-form). Results from 17 studies highlighted that the assessment condition influences the performance during the 6MWT. However, findings varied depending not only on the condition but also on the studied population, highlighting that distance traveled on the 6MWT depends on assessment condition and population type.

### 4.1. Review of the Main Findings

First, regarding the influence of different back-and-forth lengths, three of eight articles reported a significant effect [20,23,31], while four other articles did not observe significant differences, [15,21,22,26] and the eighth study observed a non-significant effect of pathway length for subjects with a lower performance (<430 m), but a significant effect for participants with higher performance (≥430 m) [24]. Few studies were conducted in a healthy population [22,25], and more than half were conducted with a population with one or more associated health conditions [15,20,21,23,24,26]. The trend within all the articles is that the longer the back and forth motion, the greater the total distance traveled on the 6MWT, which brings the score closer to the actual maximum performance of the participant [15,20,22,23,24,25]. This is explained by the fact that the greater the pathway distance, the less often the person must decelerate and turn around. In fact, a shorter back-and-forth distance calls for the subject to frequently slow down and turn 180 degrees and accelerate again during the 6MWT, which requires more time and effort, [20,21,22,23,24,25,33] leading to a shorter total distance traveled. However, the scores on the 6MWT performed with back-and-forth motion on 10 m distance were always significantly lower than in the other conditions, which means that the subject’s score is underestimated in this condition. This observation calls for a need to interpret the 6MWT score while taking into account the pathway length [20,21,25].

Seven studies had investigated the possible difference between the scores of the 6MWT performed on the ground and on a treadmill. Contrary to our second hypothesis, performing the 6MWT on a treadmill did not yield higher scores than when performed with back-and-forth motion in a ground-based hallway setting as recommended by the ATS. In fact, all of the seven studies showed that the distance traveled on a treadmill during the 6MWT was significantly lower than that travelled on the ground [8,27,28,29,30,32], except for the group of healthy elderly people in Liu et al.’s study [31]. This exception can be due to the fact that this healthy population might have not presented any muscular weakness, balance deficits, and/or anxiety compared to studies of other populations such as individuals with chronic obstructive pulmonary disease (COPD) [31]. Furthermore, according to Liu and et al. [31], using a motorized self-paced treadmill that allows natural walking speed adjustment rather than a non-motorized non-self-paced treadmill would yield scores closer to the ground-based score, although treadmill-based performance would still remain lower. It appears that both conditions cannot be considered exchangeable. In fact, even if the person does not perform any pivoting on a treadmill, many factors can contribute to a shorter traveled distance, for example, the fact that they must (1) coordinate the upper and lower limbs [32] to safely manage the treadmill [8,32] and (2) maintain balance on a moving surface [32]. These conditions make the treadmill more challenging. The observed difference in 6MWT scores according to the type of surface can also be attributed to the fact that walking on a treadmill requires more energy than walking on ground hallways [35]. In addition, despite a familiarization period, walking on a treadmill is less familiar than ground-based walking that is performed daily [28,32]. The negative impact of treadmill use on the 6MWT performance can be explained by the fact that biomechanics of walking on a treadmill are different from walking on the floor [36,37]. Indeed, on a treadmill, most participants have to manage their balance to maintain stability and keep walking on the constantly moving surface. Studies summarized in a recent systematic review by Vickery-Howe et al. (2023) have reported that while cadence and relative oxygen consumption are higher, stride length and step length are lower during treadmill walking [36]. Moreover, treadmill walking is not necessarily comfortable for all populations due to lack of control and familiarity; while it may be reliable in healthy subjects, it can appear to be challenging for people with cognitive or balance disorders, making the 6MWT performance on a treadmill less reliable. Even in healthy individuals, a learning effect of treadmill walking can interfere with the validity and reliability of the walked distance. Furthermore, walking on a treadmill may be less motivating than walking on the ground for some people, for it can be seen as more monotonous, and this can make the variability of inter-participant scores greater on a treadmill than on the ground.

Finally, three studies assessed the 6MWT in a different condition than the back-and-forth motion in the hallway [21,33,34]. Cox and et al. [34] assessed the difference between scores of the 6MWT performed on a 20 m line hallway or a 20 m oval pathway and reported similar performance during the first trial but significantly different performance during a second trial. However, on both trials, the walked distance was always better on the 20 m line pathway with back-and-forth motion than on the oval pathway. This can be explained by two reasons. First, the population studied was composed of individuals with lower extremity amputation (LEA), and the continuous turning of the oval pathway may have limited their ability to accelerate and maintain a maximum speed compared to walking on the straight pathway. Second, results may be affected because of their prothesis that limits and makes walking uncomfortable on the oval pathway [34]. According to this study, it appears preferable to use a straight-line pathway for this population in order to obtain an optimal result on the 6MWT. Fell and et al. [21] compared the 6MWT scores as recommended by the ATS (30 m) to the 6MWT performed on a 10 m back-and-forth pathway and a 10 m eight-form pathway. They reported a shorter total distance for the condition of 10 m. The results of this study showed that the short pathway affected the total distance regardless of the shape of the pathway. In Barnett and et al. [33], the results were in accordance with the observations of the above studies, which specify that the length and shape of the pathway have an impact on total distance walked during the 6MWT. In fact, the best results were yielded on a 15 m pathway, which was the longest continuous line among all the pathways in this study. Note that the eight-form was composed of two 10 m lines that cross over on a width of two meters, and the rectangular form of the pathway was six by three meters. Thus, it would be interesting to perform a study on a rectangle or an eight-form with continuous lines of at least 15 m to verify at equal pathway length the influence of performing 90-degree turns versus 180-degree turns versus continuous walking on a curve. Finally, since all the studies strictly assessed the lengths of the pathway [20,25], the results yielded on the 10 m back-and-forth pathway were lower than those obtained on the 15 m pathway, and the 10 m pathway was therefore less appropriate for the 6MWT. As for the eight-form pathway and the six-by-three meters rectangles, the authors recommend using a 15 m straight line pathway and an eight-form pathway when space is limited to better reflect the 30 m 6MWT performance as stated in the ATS guidelines.

The distribution of included studies according to health conditions does not allow for in-depth and contrasted analysis by health condition. However, the literature suggests that health conditions have an impact on performance in the 6-minute test, with individuals with health conditions performing less than healthy individuals [38,39]. Studies have also reported an age effect on the 6-min walking distance [40,41], especially when comparing young and old people, although the effect of age has been reported as minimal within groups of people of similar age ranges (e.g., between young adults) [42]. As for anthropometric factors, there is a consensus on their impact on the 6MWT performance, with studies reporting weak correlation between 6MWT and height, weight, and body mass index (BMI) [42], while other studies found significant association between the 6MWT distance and BMI [41]. Future studies are needed to elucidate the effects of demographic and anthropometric factors on performance on the 6MWT.

### 4.2. Heterogeneity Between Results in the Treadmill Adaption of the 6MWT

The significant heterogeneity observed in our meta-analysis (I² = 87%) suggests notable variability among the included studies, which could be attributed to several factors related to patient populations, study protocols, and treadmill specifications. First, the characteristics of the participant populations varied widely. Of the seven studies included, three studied healthy older adults [28,29,31], two studied patients with chronic obstructive pulmonary disease [8,31], two studied patients with cardiac conditions [27,30], and one studied patients with stable lung diseases [32]. These differences in baseline health and physical capacity likely influenced performance during the treadmill 6MWT. For example, individuals with chronic conditions may experience limitations in endurance compared to healthy participants, contributing to discrepancies in walking distances.

Second, variations in treadmill protocols were a key source of heterogeneity. For instance, Liu et al. used a treadmill combined with a virtual reality system, which could have enhanced motivation and engagement. Their study reported slightly longer distances on the treadmill compared to the traditional 6MWT in healthy older adults, although the difference was not statistically significant. This innovative setup may have introduced additional factors affecting performance that were not present in other studies using standard treadmills.

Third, differences in the initial speed settings and speed adjustment protocols added further variability. Initial treadmill speeds ranged from 0 to 3 km/h, with some studies allowing participants to personalize the speed (e.g., setting it at 50% of their chosen comfortable walking speed). These variations could have influenced participants’ pacing strategies and energy expenditure, ultimately affecting the distance covered during the test.

The high heterogeneity for this specific analysis indicates that conclusions drawn from pooled results should be interpreted cautiously, as the variability could mask underlying differences or commonalities across studies. Future research should aim for greater consistency in study protocols and consider stratifying analyses by population characteristics or treadmill settings to better understand their impact on test outcomes.

### 4.3. Limitations

First, it is possible that relevant articles in the databases were not selected because of the search equation (see Appendix A), which is possibly too specific. Although the database number may seem limited, we performed handsearching in systematic reviews and article references to overcome this limitation. In fact, two additional studies [20,31] identified through selected article references were found relevant and included in the review as they met the inclusion criteria. Second, to acquire the most information and include enough articles in the review, many conditions and populations were compared, which results in variable conclusions. It would be difficult to determine whether it was really the implementation condition alone that influenced the validity of the 6MWT results or whether other factors could have had an impact, such as the subjects’ pathology, age, instructions, or encouragements given. Third, three major databases were considered in this systematic review, namely Medline, CINAHL, and Embase. Including additional databases such as Web of Science and Scopus might have contributed to potential additional studies. However, our database selection was guided by the Cochrane Handbook [43], which specifically recommends Medline, Embase, and Cochrane’s own reference database for systematic reviews, as they are considered robust and widely accepted for high-quality evidence synthesis. Additionally, existing literature highlights substantial overlap between Scopus or Web of Science and Embase or Medline/PubMed, which supports the decision to prioritize the latter for this study.

### 4.4. Recommendations and Implications for Physiotherapy Practice

First, it should be mandatory to indicate, along with the score, the condition in which the test was performed since the test outcome may depend on the implementation conditions. This will ensure that repeated assessments can be performed in the same conditions for more comparable and valid results. Second, it would be interesting for further studies to determine which condition should be used if the 30 m pathway is not available in the clinical environment. In fact, according to current results, the closer the pathway is to 30 m, the greater the validity. Third, it would be advantageous not to use a treadmill because it underestimates the subject’s performance since it is a less familiar condition to perform the 6MWT with. Fourth, when physical constraints permit, it would be favorable to perform the hallway test with back-and-forth motion on the longest distance possible to decrease the number of 180-degree turns. Finally, results of this systematic review showed that in addition to age, height, weight, population type, and sex, which are factors used to predict the expected distance, the implementation conditions should also be considered in the score’s interpretation.

## 5. Conclusions

This systematic review presents the results of the 6MWT performed in different conditions. It was determined that the implementation condition of the 6MWT (back-and-forth length, use of treadmill, and circle, oval, or rectangle) influenced the results, and none of the conditions were exchangeable. It would be interesting to assess the courses of different shapes with at least 30 m of pathway to better determine the impact of pathway shapes in comparison with the walking path length recommended by the ATS. The continuous line on the floor without a turn and a back-and-forth distance greater than 30 m would also be interesting to study. As this review shows, we must be careful with the interpretation of the 6MWT results in clinical setting because many factors can influence the results, though further studies are needed to confirm these findings.

## Figures and Tables

**Figure 1 sensors-25-00017-f001:**
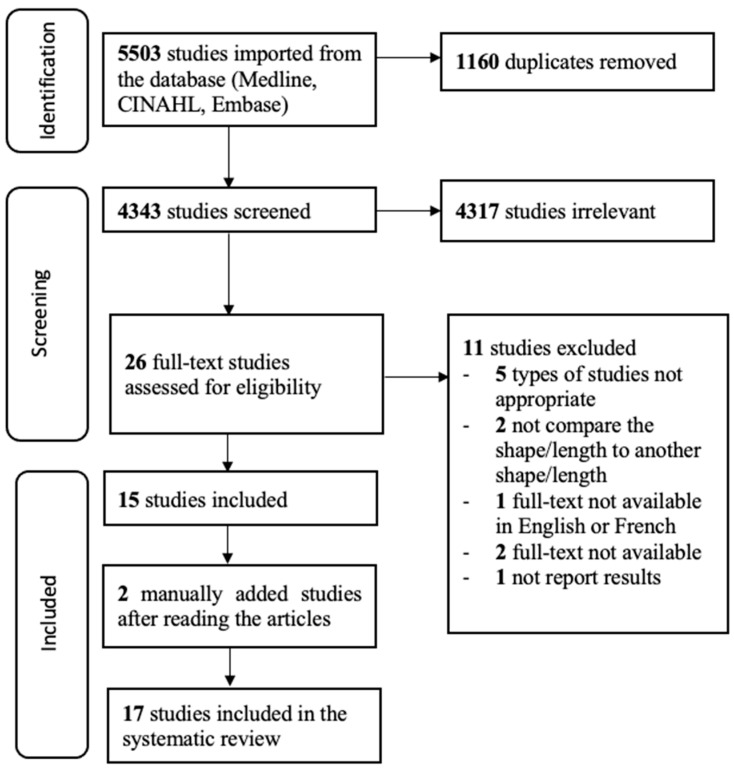
Flow chart of systematic review process.

**Figure 2 sensors-25-00017-f002:**
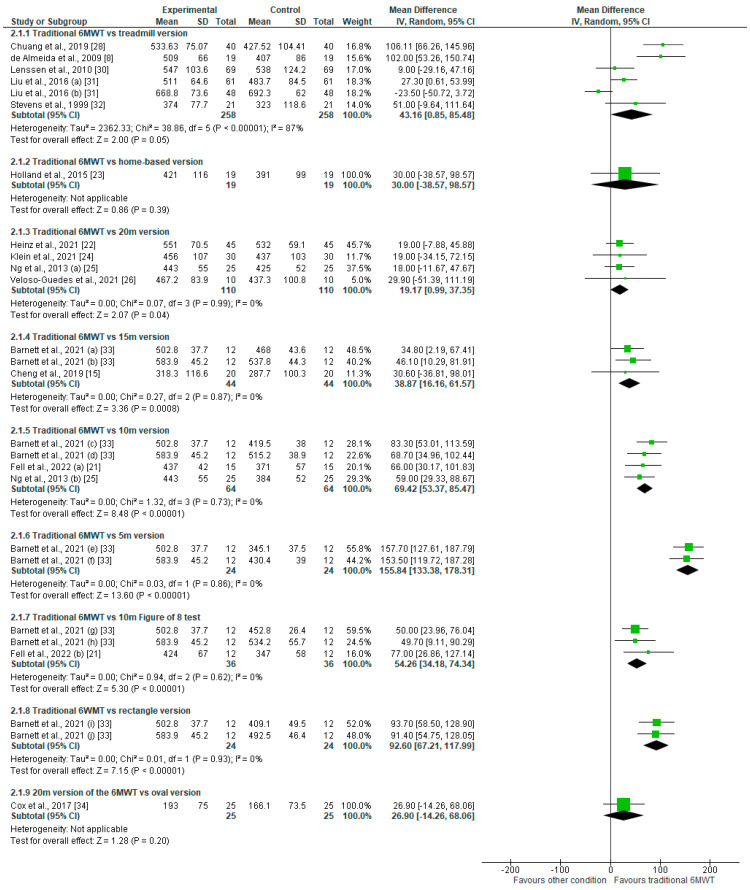
Forest plot representing the distance covered by participants in the different conditions and versions of the 6MWT [8,15,21,22,23,24,25,26,28,30,31,32,33,34].

**Table 1 sensors-25-00017-t001:** Studies characteristics.

Study	Type of Study	Assessment Context 1	Assessment Context 2	ATS Directives	Setting
Barnett C.T. et al. [33]	Not reported	Back and forth (5 m, 10 m, 15 m)	Other shape: rectangle (3 × 6 m) and the Figure 8 shape (10 m)	Y	Research
Beekman E. et al. [20]	Double randomized longitudinal experimental	Back and forth (10 m)	Back and forth (30 m)	Y	Clinical
Cheng D.K et al. [15]	Cross-sectional quantitative	Back and forth (15 m)	Back and forth (30 m)	Y	Clinical
Chuang, C.Y. et al. [28]	Comparative	Back and forth (30 m)	Motorized treadmill	Y	Research
Cox P.D. et al. [34]	Cross-sectional	Back and forth (20 m)	Oval (6 × 4 m)	Y	Clinical
De Almeida et al. [8]	Crossover	Back and forth (30 m)	Motorized treadmill	Y	Clinical
De Camargo, V.M. et al. [27]	Not reported	Back and forth (30 m)	Motorized treadmill	Y	Research
Fell B. et al. [21]	Cross-sectional	Back and forth, (10 m, 30 m)	Back and forth (30 m) and other shape (10 m Figure 8-shape)	Y	Clinical
Heinz, P.D.R. et al. [22]	Cross-sectional	Back and forth (20 m)	Back and forth (30 m)	Y	Research
Holland A.E et al. [23]	Crossover	Back and forth at home (17 m)	Back and forth hospital (30 m)	N	Clinical
Janaudis-Ferreira, T et al. [29]	Not reported	Back and forth (30 m)	Non motorized treadmill	Y	Research
Klein S.R, et al. [24]	Not reported	Back and forth (20 m)	Back and forth (30 m)	Y	Research
Lenssen A.F. et al. [30]	Pre-experimental	Rectangle (44 m)	Motorized treadmill	Y	Clinical
Liu, W.Y. et al. [31]	Longitudinal	Circle (125 m)	Motorized treadmill + VR	N	Research
Ng S.S. et al. [25]	Cross-sectional	Back and forth (10 m, 20 m)	Back and forth (30 m)	Y	Research
Stevens, D. et al. [32]	Not reported	Back and forth (30 m)	Motorized treadmill	Y	Research
Veloso-Guedes, C.A. et al. [26]	Prospective	Back and forth (20 m)	Back and forth (30 m)	Y	Research

Y: yes; N: no, VH: with virtual hallway (Gait Real-time Analysis Interactive lab; GRAIL).

**Table 2 sensors-25-00017-t002:** Participants’ characteristics.

Study	Number of Participants	Age (Average [Range])	Sex (M: W)	Presence of Associated Conditions
Barnett C.T. et al. [33]	24	Young people: 21.9 ± 1.3 [N/M] Older people: 70.2 ± 3.4 [N/M]	Young people: 12: 0 Older people: 3: 9	N
Beekman E. et al. [20]	45	67.0 [47–87]	26: 19	Y (COPD)
Cheng D.K et al. [15]	20	61.0 ± 14.0 [32–83]	14: 6	Y (post-stroke, 37 to 252 days)
Chuang, C.Y. et al. [28]	40	70.8 ± 4.9 [65–84]	17: 23	N
Cox P.D. et al. [34]	25	63.1 [N/M]	18: 7	Y (people with LEA)
De Almeida et al. [8]	19	63.4 ± 10.5 [N/M]	14: 5	Y (COPD)
De Camargo, V.M. et al. [27]	73	40.0 ± 11.0 [N/M]	20: 53	Y (PAH)
Fell B. et al. [21]	27	58.9 ± 9.6 [31–73]	9: 18	Y (cardiovascular disease, cancer, chronic respiratory disease, diabetes)
Heinz, P.D.R. et al. [22]	25	60.0 ± 10.0 [N/M]	10: 15	N
Holland A.E et al. [23]	19	71.0 ± 6.8 [N/M]]	11: 8	Y (COPD)
Janaudis-Ferreira, T et al. [29]	16	68.0 ± 3.4 [63–75]	8: 8	N
Klein S.R, et al. [24]	30	66.0 ± 8.0 [N/M]	23: 7	Y (COPD)
Lenssen A.F. et al. [30]	69	61.0 ± 16.0 [37–81]	62: 7	Y (cardiac problems)
Liu, W.Y. et al. [31]	109	COPD: 61.9 ± 6.8 [N/M] Healthy subjects: 61.6 ± 6.1 [N/M]	60: 49	Y (COPD) et N
Ng S.S. et al. [25]	25	61.0 ± 8.0 [50–70]	12: 13	N
Stevens, D. et al. [32]	21	65.0 ± 10.9 [35–79]	9: 12	Y (COPD) and N
Veloso-Guedes, C.A. et al. [26]	10	59.0 ± 10.5 [N/M]	10: 0	Y (liver cirrhosis)

M: man; W: woman; Y: yes; N: no; N/M: non mentioned; LEA: lower extremity amputation; PAH: pulmonary arterial hypertension; COPD: chronic obstructive pulmonary disease.

**Table 3 sensors-25-00017-t003:** Methodological quality of included studies according to STROBE grid.

	Title/Abstract	Introduction		Methods									Results					Discussion				Other	Total
STROBE Grid Items	1	2	3	4	5	6	7	8	9	10	11	12	13	14	15	16	17	18	19	20	21	22	
Study																							
Barnett C.T. et al. [33]	N/N	Y	Y	N	N	Y/NA	Y	N	N	N	N	Y/N/N/NA/N	N/N/N	N/N/NA	N	N/NA/NA	N	Y	Y	Y	Y	Y	10/29 34.5%
Beekman E. et al. [20]	Y/Y	Y	Y	Y	Y	Y/NA	Y	Y	N	Y	Y	Y/Y/Y/Y/N	Y/Y/N	Y/Y/Y	N	N/NA/NA	N	N	Y	Y	Y	Y	24/31 77.7%
Cheng D.K et al. [15]	Y/Y	Y	Y	Y	Y	Y/NA	Y	Y	Y	Y	Y	Y/Y/Y/Y/Y	Y/Y/Y	Y/Y/NA	Y	Y/NA/NA	Y	Y	Y	Y	Y	Y	30/30 100%
Chuang, C.Y. et al. [28]	N/Y	Y	Y	N	Y	Y/NA	Y	Y	N	Y	Y	Y/Y/N/Y/N	Y/N/N	Y/N/NA	N	N/NA/NA	N	N	N	Y	Y	N	16/30 53.3%
Cox P.D. et al. [34]	Y/Y	Y	Y	N	Y	Y/NA	Y	Y	N	Y	Y	Y/Y/N/NA/N	N/N/N	Y/N/NA	N	N/NA/NA	N	N	Y	Y	Y	N	16/29 55.2%
De Almeida et al. [8]	Y/Y	Y	N	Y	Y	Y/NA	Y	Y	Y	Y	N	Y/N/N/NA/N	N/N/N	Y/N/NA	N	N/NA/NA	N	Y	N	N	N	N	13/29 44.8%
De Camargo, V.M. et al. [27]	N/Y	Y	Y	N	Y	N/NA	Y	Y	Y	N	Y	Y/Y/N/NA/N	N/N/N	Y/N/NA	Y	N/NA/NA	N	Y	N	N	N	N	13/29 44.8%
Fell B. et al. [21]	Y/Y	Y	N	Y	Y	Y/NA	Y	Y	Y	N	Y	Y/Y/N/Y/N	Y/Y/N	Y/Y/NA	Y	Y/NA/NA	Y	Y	Y	Y	N	Y	24/31 77.74%
Heinz, P.D.R. et al. [22]	N/Y	Y	Y	Y	N	Y/NA	Y	Y	Y	Y	N	Y/N/N/ NA/N	Y/Y/N	Y/Y/NA	Y	N/NA/NA	N	Y	Y	Y	Y	N	19/29 65.5%
Holland A.E et al. [23]	N/Y	Y	Y	Y	Y	Y/NA	Y	Y	N	Y	N	Y/N/N/NA/N	N/N/N	Y/N/NA	N	N/NA/NA	N	Y	Y	Y	Y	N	15/29 51.7%
Janaudis-Ferreira, T et al. [29]	N/Y	Y	Y	N	Y	N/NA	Y	Y	Y	Y	N	Y/N/N/NA/N	N/N/N	Y/N/NA	N	N/NA/NA	N	Y	N	Y	Y	Y	14/29 48.3%
Klein S.R, et al. [24]	N/Y	Y	Y	Y	Y	Y/NA	Y	Y	N	Y	Y	Y/Y/N/NA/N	N/N/N	Y/Y/NA	Y	N/NA/NA	Y	Y	Y	N	Y	Y	20/29 69.0%
Lenssen A.F. et al. [30]	Y/Y	Y	Y	N	Y	Y/NA	Y	Y	Y	Y	N	Y/N/N/NA/N	Y/Y/N	Y/N/NA	Y	N/NA/NA	N	Y	Y	Y	Y	N	19/29 65.5%
Liu, W.Y. et al. [31]	N/Y	Y	Y	Y	Y	Y/NA	Y	Y	N	Y	N	Y/N/N/NA/N	N/N/N	Y/Y/NA	Y	N/NA/NA	N	Y	Y	Y	Y	Y	18/29 62.1%
Ng S.S. et al. [25]	Y/Y	Y	Y	Y	N	Y/NA	Y	Y	Y	Y	N	Y/N/N/NA/N	N/N/N	Y/N/NA	Y	N/NA/NA	N	Y	Y	Y	Y	Y	18/29 62.1%
Stevens, D. et al. [32]	N/Y	Y	Y	N	Y	Y/NA	Y	Y	N	N	N	Y/N/N/NA/N	N/N/N	Y/N/NA	Y	N/NA/NA	N	Y	N	Y	N	N	12/29 41.4%
Veloso-Guedes, C.A. et al. [26]	Y/Y	Y	Y	N	Y	Y/NA	Y	Y	N	N	N	Y/N/N/NA/N	N/N/N	Y/N/NA	N	N/NA/NA	N	Y	N	N	N	N	11/29 37.9%

Y: yes; N: no; NA: non applicable.

**Table 4 sensors-25-00017-t004:** Results obtained at the 6MWT and conclusion.

Study	Barnett C.T. et al. [33]	Beekman E. et al. [20]	Cheng D.K et al. [15]	Chuang, C.Y. et al. [28]	De Almeida et al. [8]
Total distance travelled in the 6MWT	Youth: BF (5 m): 430.4 m BF (10 m): 515.2 m BF (15 m): 537.8 m Rectangle (3 m × 6 m): 492.5 m 8-form (10 m): 534.2 m	Average difference between the best trial with AR (10 m) vs. BF (30 m): −49.5 m	BF (15 m): 278.3 m ± 99.9 m BF (30 m): 294.9 m ± 107.5 m	Trial 2: BF (30 m): 533.63 m ± 75.07 m Motorized TR: 427.52 m ± 104.41 m	BF (20 m): 193.0 m Oval (6 m × 4 m): 166.1 m
	E: BF (5 m): 345.1 m BF (10 m): 419.5 m BF (15 m): 468.0 m Rectangle (3 m × 6 m): 409.0 m 8-form (10 m): 452.8 m			Trial 1: BF (30 m): 521.86 m ± 67.63 m Motorized TR: 401.95 ± 104.60 m	BF (20 m): 173.3 m Oval (6 m × 4 m): 160.3 m BF (30 m): 509 m ± 66 m Motorized TR: 407 m ± 86 m
Conclusion	The hallways (15 m) and the eight-form produced the best distance on the 6MWT compared to results from the 10 m hallway and the rectangle (3 × 6 m), and from the 5 m hallway. The recommendations were the usage of a continuous course with a maximal straight line.	Results were significantly better in the hallway (30 m) than the hallway (10 m). The conditions were not exchangeable.	Best results in the hallway (30 m) than in the hallway (15 m); Great accuracy for the hallway (15 m) and great correlation with the hallway (30 m). Recommendation to perform in the hallway (15 m and more).	Results were significantly better in the hallway (30 m) relative to the motorized treadmill. Conditions were non-exchangeable.	Both conditions were valid but non-exchangeable. Conditions non-exchangeable.
Study	Cox P.D. et al. [34]	De Camargo, V.M. et al. [27]	Heinz, P.D.R. et al. [22]	Holland A.E et al. [23]	Janaudis-Ferreira, T et al. [29]
Total distance travelled in the 6MWT	Trial 2: BF (20 m): 193.0 m Oval (6 m × 4 m): 166.1 m	BF (30 m): 436 m ± 127 m Motorized TR: 372 m ± 121 m	Trial 2: BF (20 m): 532.0 m ± 59.1 m BF (30 m): 551.0 m ± 70.5 m	BF hospital (30 m): 421 m ± 116 m Home (17 m [7–45] m): 391 m ± 99 m	BF (30 m): [540–561] m Non motorized TR: [376–408] m
	Trial 1: BF (20 m): 173.3 m Oval (6 m × 4 m): 160.3 m		Trial 1: BF (20 m): 533.0 m ± 73.1 m BF (30 m): 544.0 m ± 72.1 m		
Conclusion	There were no significant differences between the two conditions on day one of the trial, but results were significantly better in the hallway (20 m) than on the oval pathway (5 m · 4 m) during the second day of the trial.	Better results in the hallway (30 m) than on the motorized treadmill. Conditions were non-exchangeable.	Best results in the hallway (30 m) than in the hallway (20 m), but the difference is not clinically significant.	Results significantly better in 30 m hallway than 17 m hallway.	Results significantly better in the hallway (30 m) than on the non motorized treadmill. Conditions were non exchangeable.
Study	Klein S.R, et al. [24]	Lenssen A.F. et al. [30]	Liu, W.Y. et al. [31]		Ng S.S. et al. [25]
Total distance travelled in the 6MWT	Trial 2: BF (20 m): 437 m ± 103 m BF (30 m): 456 m ± 107 m	Rectangle (44 m): 547.0 m ± 103.6 m Motorized TR: 538.0 m ± 124.2 m	COPD: Circle (125 m): 481.4 m and 506.2 m Motorized TR with integrated: 449.7 m and 476.4 m		Rotation on the dominant side: BF (10 m): 377 m BF (20 m): 428 m BF (30 m): 456 m
	Trial 1: BF (20 m): 429 m ± 97 m BF (30 m): 444 m ± 115 m		Healthy subject: Circle (125 m): 668.8 m Motorized TR with integrated: 639.9 m and 689.5 m		Rotation on the dominant side: BF (10 m): 384 m BF (20 m): 425 m BF (30 m): 443 m
Conclusion	Better results in the hallway (30 m). A difference of 30 m or more in the distance walked in hallway of 30 m compared to hallway of 20 m showed by the cutoff point (ROC curve) of 430 m.	Better results on a rectangular course of 44 m than on the motorized TR. Conditions were non exchangeable.	Results were significantly better for the 125 m circular course for COPD patients. Results were significantly better on the motorized treadmill with integrated virtual reality for the healthy subjects.		Results were better on longer distances (20 m and 30 m) than in the hallway 10 m. Condition not exchangeable and no difference observed because of pivoting.
Study	Stevens, D. et al. [32]	Veloso-Guedes, C.A. et al. [26]	Fell B. et al. [21]		
Total distance travelled in the 6MWT	F (30 m): 374.9 m ± 77.7 m Motorized TR: 323.1 m ± 118.6 m	BF (20 m): 437.3 m ± 100.8 m BF (30 m): 464.8 m ± 80.0 m	Group B BF (10 m figure eight): 347 m ± 58 m BF (30 m): 424 m ± 67 m		
			Group A BF (10 m): 371 m ± 57 m BF (30 m): 437 m ± 42 m		
Conclusion	Results significantly better in the hallway (30 m) than on the motorized TR. Condition non exchangeable.	Results were better in the 30 m-hallway than in the 20 m-hallway, but the difference is not clinically significant.	Results were greater on the standard pathway (30 m) than alternative pathways (10 m and 10 m figure eight), however, the difference was not statistically significant		

Meters (m), back and forth (BF), treadmill (TR), virtual reality (VR), receiver operating characteristic (ROC).

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
