# Peer review of "Impact of Pathway Shape and Length on the Validity of the 6-Minute Walking Test: A Systematic Review and Meta-Analysis"

_sensors, 2024, doi:10.3390/s25010017_

Round 1

Reviewer 1 Report

Comments and Suggestions for Authors

First, I wish to congratulate the authors of the paper "Impact of pathway shape and length on the validity of the 6-minute walking test: a systematic review and meta-analysis" for their correct methodologically review. My main concern is that there is a lack of the interpretation of the findings from a physiology/biomechanics point of view. The authors should work more on the discussion of the results highlighting the changes occuring when the length varies with an emphasis on the additional muscular work. 

Author Response

Comment: First, I wish to congratulate the authors of the paper "Impact of pathway shape and length on the validity of the 6-minute walking test: a systematic review and meta-analysis" for their correct methodologically review. My main concern is that there is a lack of the interpretation of the findings from a physiology/biomechanics point of view. The authors should work more on the discussion of the results highlighting the changes occuring when the length varies with an emphasis on the additional muscular work.

Response: We thank the reviewer for this comment. We have now extended the discussion section to further explore potential factors that can explain the observed difference. We have added the following paragraph in the discussion section:

The negative impact of treadmill use on the 6MWT performance can be explained by the fact that biomechanics of walking on treadmill are different from walking on the floor [36, 37]. Indeed, on treadmill, most of participants have to manage for their balance to maintain stability and keep walking on the constantly moving surface. Studies summarized in a recent systematic review by Vickery-Howe et al. (2023) have reported that while cadence and relative oxygen consumption are higher, stride length and step length are lower during treadmill walking [36]. Moreover, treadmill walking is not necessarily comfortable for all populations due to lack of control and familiarity; while it may be reliable in healthy subjects, it can appear to be challenging for people with cognitive or balance disorders, making the 6MWT performance on treadmill less reliable). Even in healthy individuals, a learning effect of treadmill walking can interfere with the validity and the reliability of the walked distance. Furthermore, walking on a treadmill may be less motivating than walking on the ground for some people for it can be seen as more monotonous, and this can make the variability of inter-participant scores greater on treadmill than on the ground.

Thank you for your comments and for the time dedicated to review our manuscript.

Reviewer 2 Report

Comments and Suggestions for Authors

The manuscript reviews the impact of pathway shape, length, and surface conditions on the validity of the six-minute walking test (6MWT). However, this review paper does not discuss anything related to sensors. Therefore, the content of this paper does not align with the journal’s scope. It is recommended to submit this paper to another relevant journal. Besides, here are some comments for the paper:

  1. During the article selection stage, it does need to mention how many people are engaged. Because readers’ focus is on how many relevant references are chosen.
  2. In this paper, the references come from three database: Medline, CINAHL, and Embase. It is recommended to include Web of Science and Scopus into the database because these two databases include more comprehensive and prestigious references.
  3. In the conclusion part, it said that “further studies are needed to provide clear evidence”. What does the “clear evidence” mean?
  4. The one section of Table 3 (line 190) has a different table format compared to other parts of the Table 3. The format of the table shall be consistent. The same issue occurs on Table 4, which should also be fixed.
  5. What does the screenshot in line 232 stand for? Moreover, is it a figure or a table?

Author Response

We wish to express our gratitude to the reviewer for his/her constructive comments. Please find below our point by point responses to reviewer 2’s comments, as well as track changes in the re-submitted files.

Comment 1: During the article selection stage, it does need to mention how many people are engaged. Because readers’ focus is on how many relevant references are chosen.

Response 1 : We thank the Reviewer for his/her contributive comments. While we agree that the Results section underlines the number of references included during each step, we would like to draw the Reviewer’s attention on the fact that the number of people engaged is presented in the Methods section (as recommended by PRISMA guidelines). In fact, in section 2.3, pages 2 and 3, lines 90 to 99, we mentioned the number of people involved in articles selection, as follows: “…First, two reviewers initially screened the articles by reading only the title and the abstract to determine whether they were potentially eligible. In the absence of consensus, a third reviewer was called upon to participate. […] In the case of a conflict, the third reviewer was involved in the final decision of inclusion or exclusion. In addition, references of selected articles were checked for additional articles to be included.    

Comment 2: In this paper, the references come from three database: Medline, CINAHL, and Embase. It is recommended to include Web of Science and Scopus into the database because these two databases include more comprehensive and prestigious references.

Response 2: We agree with this comment. However, while we recognize the value of these additional databases, we respectfully note that the 5-day deadline for revisions is too short to conduct additional searches and integrate comprehensive findings from these two additional databases. Furthermore, our database selection was guided by the Cochrane Handbook, which specifically recommends Medline, Embase, and Cochrane's own reference database for systematic reviews, as they are considered robust and widely accepted for high-quality evidence synthesis. Additionally, existing literature highlights substantial overlap between Scopus/Web of Science and Embase/PubMed, which supports our decision to prioritize the latter for this study.

For the above-mentioned reasons, we believe that the three databases that were searched to retrieve articles cover the relevant literature pertaining to this systematic review. However, we acknowledge the non-inclusion of Scopus and Web of Science, as a potential limitation of our methodology and have included this point in the manuscript's discussion section. Please, see the Limitation section, where the following paragraph is added:

Third, three major databases were considered in this systematic review, namely Medline, CINAHL and Embase. Including additional databases such as Web of Science and Scopus might have contributed to potential additional studies. However, our database selection was guided by the Cochrane Handbook [38], which specifically recommends Medline, Embase, and Cochrane's own reference database for systematic reviews, as they are considered robust and widely accepted for high-quality evidence synthesis. Additionally, existing literature highlights substantial overlap between Scopus or Web of Science and Embase or Medline/PubMed, which supports the decision to prioritize the latter for this study.”

Comment 3: In the conclusion part, it said that “further studies are needed to provide clear evidence”. What does the “clear evidence” mean?

Response 3: By “clear evidence” we mean robust evidence. For clarity, we have reworded the sentence as follows: “...further studies are needed to confirm these findings”.

Comment 4: The one section of Table 3 (line 190) has a different table format compared to other parts of the Table 3. The format of the table shall be consistent. The same issue occurs on Table 4, which should also be fixed.   Response 4: We thank the reviewer for this comment. We have modified the Tables in the revised manuscript; and the format is now consistent throughout. See Table 3 and Table 4 in the revised manuscript.    

Comment 5: What does the screenshot in line 232 stand for? Moreover, is it a figure or a table?

Response 5: This is a figure. We thank the reviewer for this comment. Indeed, we have omitted to indicate the title of this figure in the initial submission. We have now fixed it and the screen short is titled:

Figure 2: Forest-plot representing the distance covered by participants in the different conditions and versions of the 6MWT.

Thank you for your comments and for the time dedicated to review our manuscript.

Reviewer 3 Report

Comments and Suggestions for Authors

This systematic review examines how different implementation conditions—such as the length of the back-and-forth walking path, the use of a treadmill, and variations in pathway shapes (e.g., circle, oval, rectangle)—affect the performance and validity of the Six-Minute Walk Test (6MWT). The review analyzes 17 studies and concludes that the 6MWT conditions influence the test results, with significant variations observed across different populations.

Major Comments

1.       The inclusion and exclusion criteria were well-defined, but it would be helpful if the authors provided more specific justification for excluding certain studies. For example, what led to the decision to exclude studies with particular health conditions or those with smaller sample sizes?

2.       The significant heterogeneity observed in the meta-analysis (I² = 87%) was noted but not thoroughly discussed. The review should provide a more detailed explanation of why such high heterogeneity exists and how it may influence the conclusions. Some potential sources of heterogeneity (e.g., variations in patient population study protocols) could have been explored in greater depth.

3.       The review mentions that different populations (e.g., healthy individuals vs. those with health conditions) were included, but it does not fully address how these factors may influence the results. A deeper analysis of how health status, age, or other demographic factors may interact with test conditions would strengthen the conclusions.

4.       While the review recommends using a 30-meter back-and-forth distance as ideal, it could be more specific in explaining how clinicians can adapt the test in different clinical settings where such conditions might not be feasible. More practical guidance on maintaining test validity under constrained conditions (e.g., limited space) would be beneficial.

5.       The negative impact of treadmill use on the 6MWT performance is discussed, but the review could benefit from further exploration of why treadmill walking is less reliable. The interaction between the treadmill's mechanical nature and physiological responses in different populations should be explored in more detail to explain the discrepancies in results more clearly.

Minor Comments

1.       While the authors discuss the potential limitations of the included studies, it would be helpful if they had provided a formal assessment of the quality or risk of bias of the studies. Incorporating a tool like the Cochrane Risk of Bias tool would enhance the credibility of the findings.

2.       The review uses terms like "MD" and "p-values" without providing enough explanation for readers who may not be familiar with statistical analysis. A brief definition of key statistical terms would improve accessibility, especially for clinicians and non-experts.

3.       The review calls for further research on various aspects of the 6MWT, but it could be more specific in identifying the most pressing gaps. For example, future studies could focus on the impact of different treadmill types (motorized vs. non-motorized) or further investigate the influence of specific patient conditions (e.g., COPD, amputation) on 6MWT outcomes.

Author Response

We wish to express our gratitude to the reviewer for his/her constructive comments. Please find below our point by point responses to reviewer 3’s comments, as well as track changes in the re-submitted files.

Comment 1: The inclusion and exclusion criteria were well-defined, but it would be helpful if the authors provided more specific justification for excluding certain studies. For example, what led to the decision to exclude studies with particular health conditions or those with smaller sample sizes?

Response 1: We thank the reviewer for this comment. We did not exclude studies based on participants' health status or sample size. In fact, in section 2.2, page 2, lines 84 to 88, it is mentioned that only abstract, systematic review and meta-analyses were excluded during studies screening and selection. Furthermore, in figure 1, page 4, line 145, the 11 studies excluded were not based on participants' health status or sample size.

Comment 2: The significant heterogeneity observed in the meta-analysis (I² = 87%) was noted but not thoroughly discussed. The review should provide a more detailed explanation of why such high heterogeneity exists and how it may influence the conclusions. Some potential sources of heterogeneity (e.g., variations in patient population study protocols) could have been explored in greater depth.

Response 2: We appreciate the reviewer’s suggestion to address the significant heterogeneity observed in the meta-analysis (I² = 87%). In response, we have added a new section to the manuscript (4.2. Heterogeneity between Results in the Treadmill Adaptation of the 6MWT), which explores the potential sources of this heterogeneity in greater depth.

In this section, we discuss how variations in participant populations, study protocols, and treadmill specifications contributed to the observed heterogeneity. Specifically, we note that the included studies involved diverse populations, such as healthy older adults, patients with COPD, cardiac conditions, or stable lung diseases, which likely influenced performance during the treadmill 6MWT. Additionally, differences in treadmill setup and protocols, such as the use of a virtual reality-enhanced treadmill in one study and variations in initial speed settings (ranging from 0 to 3 km/h, with some studies allowing personalized speeds), further contributed to variability in outcomes. These methodological and population differences underscore the challenges in standardizing the treadmill adaptation of the 6MWT.

The new section 4.2. is made of the following paragraphs:

The significant heterogeneity observed in our meta-analysis (I² = 87%) suggests notable variability among the included studies, which could be attributed to several factors re-lated to patient populations, study protocols, and treadmill specifications. First, the characteristics of the participant populations varied widely. Of the seven studies included, three included on healthy older adults [28, 29, 31], two on patients with chronic obstructive pulmonary disease [8, 31], two on patients with cardiac conditions [27, 30], and one on patients with stable lung diseases [32]. These differences in baseline health and physical capacity likely influenced performance during the treadmill 6MWT. For example, individuals with chronic conditions may experience limitations in endurance compared to healthy participants, contributing to discrepancies in walking distances.

Second, variations in treadmill protocols were a key source of heterogeneity. For instance, Liu et al. used a treadmill combined with a virtual reality system, which could have enhanced motivation and engagement. Their study reported slightly longer distances on the treadmill compared to the traditional 6MWT in healthy older adults, although the difference was not statistically significant. This innovative setup may have introduced additional factors affecting performance that were not present in other studies using standard treadmills.

Third, differences in the initial speed settings and speed adjustment protocols added further variability. Initial treadmill speeds ranged from 0 to 3 km/h, with some studies allowing participants to personalize the speed (e.g., setting it at 50% of their chosen comfortable walking speed). These variations could have influenced participants’ pacing strategies and energy expenditure, ultimately affecting the distance covered during the test.

The high heterogeneity for this specific analysis indicates that conclusions drawn from pooled results should be interpreted cautiously, as the variability could mask underlying differences or commonalities across studies. Future research should aim for greater consistency in study protocols and consider stratifying analyses by population characteristics or treadmill settings to better understand their impact on test outcomes.

Comment 3: The review mentions that different populations (e.g., healthy individuals vs. those with health conditions) were included, but it does not fully address how these factors may influence the results. A deeper analysis of how health status, age, or other demographic factors may interact with test conditions would strengthen the conclusions.

Response 3: Thank you for this comment. We agree that health status, age and other demographic factors such as height, weight can interact with test conditions. However, it is recommended that a systematic review be focused on a limited number of straightforward questions to increase the sensitivity of literature search and to facilitate reaching meaningful conclusions. The present work focuses on the impact of pathway shape and length on the validity of the 6MWT. Assessing health status, age, other demographic factors and test conditions in the same review would lead to too much information, making it difficult to read. We think that these other factors could be addressed in further studies. However, we have now included in the Discussion, the following paragraph on the potential influence of these factors on the 6MWT performance (Page 17; lines 340-351):

The distribution of included studies according to health conditions does not allow for in-depth and contrasted analysis by health condition. However, the literature suggests that health conditions have an impact on performance in the 6-minute test, with individuals with health conditions performing less than healthy individuals [38, 39]. Studies have also reported an age effect on the 6-minute walking distance [40, 41], especially when comparing young and old people, although the effect of age had been reported as minimal within groups of people of similar age ranges (e.g. between young adults) [42]. As for anthropometric factors, there is a consensus on their impact of the 6MWT performance, with studies reporting weak correlation between 6MWT and height, weight and body mass index (BMI) [42], while other studies found significant association between the 6MWT distance and BMI [41…]. Future studies are needed to elucidate the effects of demographic and anthropometric factors on performance in the 6MWT.

Comment 4: While the review recommends using a 30-meter back-and-forth distance as ideal, it could be more specific in explaining how clinicians can adapt the test in different clinical settings where such conditions might not be feasible. More practical guidance on maintaining test validity under constrained conditions (e.g., limited space) would be beneficial.

Response 4: In section 4.1, page 17, line 309 to 312, some authors recommend using 15 meters straight line pathway and eight-form pathway when space is limited to better reflect the 30 meters 6MWT performance as stated in the ATS guidelines.

Comment 5: The negative impact of treadmill use on the 6MWT performance is discussed, but the review could benefit from further exploration of why treadmill walking is less reliable. The interaction between the treadmill's mechanical nature and physiological responses in different populations should be explored in more detail to explain the discrepancies in results more clearly.

Response 5: We thank the reviewer for this comment. We have now extended the discussion section to further explore potential factors that can explain the observed difference. We have added the following paragraph in the discussion section (Page 16; lines 294-308):

The negative impact of treadmill use on the 6MWT performance can be explained by the fact that biomechanics of walking on treadmill are different from walking on the floor [36, 37]. Indeed, on treadmill, most of participants have to manage for their balance to maintain stability and keep walking on the constantly moving surface. Studies summarized in a recent systematic review by Vickery-Howe et al. (2023) have reported that while cadence and relative oxygen consumption are higher, stride length and step length are lower during treadmill walking [36]. Moreover, treadmill walking is not necessarily comfortable for all populations due to lack of control and familiarity; while it may be reliable in healthy subjects, it can appear to be challenging for people with cognitive or balance disorders, making the 6MWT performance on treadmill less reliable). Even in healthy individuals, a learning effect of treadmill walking can interfere with the validity and the reliability of the walked distance. Furthermore, walking on a treadmill may be less motivating than walking on the ground for some people for it can be seen as more monotonous, and this can make the variability of inter-participant scores greater on treadmill than on the ground.”

Comment 6: While the authors discuss the potential limitations of the included studies, it would be helpful if they had provided a formal assessment of the quality or risk of bias of the studies. Incorporating a tool like the Cochrane Risk of Bias tool would enhance the credibility of the findings.

Response 6: The methodological quality of the articles included in this review was assessed with the Strengthening the Reporting of Observational Studies in Epidemiology (STROBE) tool. The methodological quality scores of included studies are presented in Table 3.

Comment 7: The review uses terms like "MD" and "p-values" without providing enough explanation for readers who may not be familiar with statistical analysis. A brief definition of key statistical terms would improve accessibility, especially for clinicians and non-experts.

Response 7: MD means “mean difference”, and statistical significance is defined as a p-value < 0.05. The following paragraphs are now added in the Methods section to clarify this point :

Page 3, lines 132 and 137: According to the Cochrane Handbook, “the MD is a standard statistic that measures the absolute difference between the mean value in two groups or conditions in a trial. It estimates the amount by which the experimental intervention/condition changes the outcome on average compared with the control/other condition. It can be used as a summary statistic in meta-analysis when outcome measurements in all studies are made on the same scale.”

Page 3, lines 132 and 137: Statistical significance was defined as a p-value < 0.05. The p-value, ranging from 0 to 1, is the probability of observing a result as extreme as, or more extreme than, the observed one, assuming the null hypothesis is true. It is directly related to the risk of making a Type I error, which occurs when the null hypothesis is incorrectly rejected.

Comment 8: The review calls for further research on various aspects of the 6MWT, but it could be more specific in identifying the most pressing gaps. For example, future studies could focus on the impact of different treadmill types (motorized vs. non-motorized) or further investigate the influence of specific patient conditions (e.g., COPD, amputation) on 6MWT outcomes.

Response 8: To address this comment as well as some other comments from reviewers, we have added the following paragraphs:

Page 17; lines 340-351: The distribution of included studies according to health conditions does not allow for in-depth and contrasted analysis by health condition. However, the literature suggests that health conditions have an impact on performance in the 6-minute test, with individuals with health conditions performing less than healthy individuals [38, 39]. Studies have also reported an age effect on the 6-minute walking distance [40, 41], especially when comparing young and old people, although the effect of age had been reported as minimal within groups of people of similar age ranges (e.g. between young adults) [42]. As for anthropometric factors, there is a consensus on their impact of the 6MWT performance, with studies reporting weak correlation between 6MWT and height, weight and body mass index (BMI) [42], while other studies found significant association between the 6MWT distance and BMI [41…]. Future studies are needed to elucidate the effects of demographic and anthropometric factors on performance in the 6MWT.

Page 18; lines 378-382: The high heterogeneity for this specific analysis indicates that conclusions drawn from pooled results should be interpreted cautiously, as the variability could mask un-derlying differences or commonalities across studies. Future research should aim for greater consistency in study protocols and consider stratifying analyses by population characteristics or treadmill settings to better understand their impact on test outcomes.

We thank the reviewer for these very helpful comments and for the time dedicated to review our manuscript.
